# Evaluation of a Ten-Antigen Immunodot Test in Autoimmune Hepatitis and Primary Biliary Cholangitis: Lessons Learned for a Tertiary Care Academic Hospital

**DOI:** 10.3390/diagnostics14171882

**Published:** 2024-08-28

**Authors:** Giulia Zorzi, Perrin Ngougni Pokem, Geraldine Dahlqvist, Bénédicte Délire, Nicolas Lanthier, Peter Starkel, Yves Horsmans, Cedric Aupaix, Samia Jnaoui, Damien Gruson

**Affiliations:** 1Department of Laboratory Medicine, Cliniques Universitaires Saint-Luc, UCLouvain, Avenue Hippocrate 10, B-1200 Brussels, Belgium; perrin.ngougni@uclouvain.be (P.N.P.); cedric.aupaix@saintluc.uclouvain.be (C.A.); samia.jnaoui@saintluc.uclouvain.be (S.J.); damien.gruson@saintluc.uclouvain.be (D.G.); 2Department of Hepato-Gastroenterology, Cliniques Universitaires Saint-Luc, UCLouvain, Avenue Hippocrate 10, B-1200 Brussels, Belgium; geraldine.dahlqvist@saintluc.uclouvain.be (G.D.); benedicte.delire@saintluc.uclouvain.be (B.D.); nicolas.lanthier@saintluc.uclouvain.be (N.L.); peter.starkel@saintluc.uclouvain.be (P.S.); yves.horsmans@saintluc.uclouvain.be (Y.H.); 3Endocrinology, Diabetes and Nutrition Research Centre, Institute for Experimental and Clinical Research, Cliniques Universitaires St-Luc and UCLouvain, Avenue Hippocrate 10, B-1200 Brussels, Belgium

**Keywords:** immunodot, antigens, autoantibodies, autoimmune hepatitis, primary biliary cholangitis, diagnostics

## Abstract

Autoimmune diseases of the liver and biliary tract require timely and accurate diagnosis. This study evaluates the D-tek panel (D-Tek, Mons, Belgium) of 10 immunodot antigens for its effectiveness in diagnosing autoimmune hepatitis (AIH) and primary biliary cholangitis (PBC). We retrospectively analysed serum samples from 111 patients who had undergone routine testing, including indirect immunofluorescence (IIF) and enzyme-linked immunosorbent assays (ELISA), to confirm or exclude autoimmune liver or biliary tract disease. The panel tested for M2/nPDC, M2/OGDC-E2, M2/BCOADC-E2, M2/PDC-E2, gp210, sp100, LKM1, LC1, SLA, and F-actin antigens. Results showed that all positive IIF+ELISA results were confirmed by the immunodot panel, except for two samples from patients who had never been diagnosed with AIH. The immunodot test identified over 20 additional autoantibodies in samples initially negative by IIF, corroborated by laboratory imaging and medical history. The immunodot technique proved to be a quick, sensitive, and specific method with high overall accuracy. This study suggests that the immunodot technique may be an effective screening and confirmatory method for autoimmune liver diseases, potentially improving diagnostic efficiency and accuracy in clinical practice.

## 1. Introduction

Autoimmune hepatitis (AIH) is a chronic liver disease of unknown aetiology that preferentially affects women and is characterized by interface hepatitis on liver histology, hypergammaglobulinemia, and circulating autoantibodies and a favourable response to immunosuppression [1]. AIH is subdivided into type 1 and type 2: AIH-1 is more common and affects both children and adults, whereas AIH-2 is mainly a paediatric disease. Interestingly, an increasing number of patients are also diagnosed at older ages. Some authors identify a third type, AIH-3, characterized by the presence of specific autoantibodies and a more severe course, while for others, it is only a variant of AIH-1 [1,2]. The prevalence of AIH varies from 160 to 250 per 1 million in Europe and North America. Even if the prevalence and incidence data are limited, the last 10 years have seen an increase in AIH, not only in Northern Europe but also in Mediterranean countries [1,2]. Primary biliary cholangitis (PBC), previously named primary biliary cirrhosis, is a chronic cholestatic liver disease characterized by a specific bile duct pathology with progressive intrahepatic duct destruction leading to cholestasis. PBC is potentially fatal and can have both intrahepatic and extrahepatic complications [3]. Its prevalence is estimated to be between 19 and 42 cases per million inhabitants, with higher peaks in the north of England. The incidence is also increasing, especially in Europe and North America [1,4].

The early diagnosis is critical because, if untreated, these diseases can progress to liver cirrhosis and death from liver failure [2,5]. The detection of autoimmune liver disease-related antibodies is a prerequisite for the diagnosis of AIH and PBC and is part of the diagnostic scoring system for these pathologies [6]. Type 1 AIH is associated with F-actin-reactive smooth muscle autoantibodies (SMAs) and autoantibodies to soluble liver antigen (SLA/LP). Type 2 autoimmune hepatitis is associated with liver kidney microsome (LKM-1) and liver cytosol (LC-1) autoantibodies [1]. Anti-LKM1 are considered more frequent in European patients and are typically unaccompanied by SMA [7]. PBC is associated with a mitochondria-associated autoantibody (AMA) and PBC-specific antinuclear autoantibodies. These latter include antibodies to the nuclear pore complex (NPC) that target gp210 (a glycoprotein of the nuclear pore complex) and nucleoporin p62, as well as antibodies to multiple nuclear dots that target Sp100 (a nuclear body speckled 100KDa protein) and PML (promyelocytic leukemia protein), which are found in about 50% of patients with PBC [5]. The reference method for testing liver-related autoantibodies is still indirect immunofluorescence (IIF) on triple rodent tissue, i.e., liver, kidney, and stomach. This technique potentially allows the simultaneous detection of the main liver-related autoantibodies, including antinuclear antibodies (ANA), SMA, anti-LKM1, anti-LC1, and AMA [8]. For the detection of anti-sp100 antibodies manifesting in a nuclear dot pattern on Hep2 cells) and anti gp210 (nuclear membrane pattern), ANA is not specific and could be detected in rheumatological diseases or in hepatitis of other origins [6]. It must be emphasized that the IIF method requires trained laboratory staff, is observer-dependent, and remains poorly standardized. Moreover, the quality of the substrates differs among laboratories/manufacturers and over time. Additionally, the detection of these antibodies must be confirmed using an antigen-specific technique, such as ELISA or dot-blot, among others [6].

In Europe, there is consensus that solid-phase assays such as enzyme-linked immunosorbent assays (ELISA) or immunoblots should only be used to confirm the results of IIF but not for initial screening. However, in the USA, for instance, ELISA is frequently used for screening purposes [6]. For the diagnostic workup of autoimmune liver diseases, anti-soluble anti-SLA/LP should already initially be tested using ELISA or immunoblot since these autoantibodies cannot be detected via IIF and have high specificity for AIH [6].

As for the ELISA, in our experience, since these diseases are rather rare, the confirmatory or additional tests force our laboratory, from a practical point of view, to accumulate several samples for batch analysis in order to avoid wasting controls and calibrations, which would increase costs. This could also clearly delay result delivery to physicians and then diagnosis and treatment.

Therefore, in our laboratory, if there is a suspicion of autoimmune pathology of the liver or biliary tract, we perform screening with IIF (AMA, ASMA, or LKM according to the clinician’s request) and, in the case of a positive result, we confirm it with an ELISA method. If the screening is negative but the clinical suspicion remains important, the laboratory, in agreement with the clinician, decides whether to expand the panel and add ELISA tests (e.g., in the case of anti-SLA) or ANA (if not requested). We do not have tests to confirm anti-sp100 and gp210 antibodies, so we send samples to an external laboratory in case of a specific request or a positive screening with a likely ANA pattern and clinical suspicion.

There is, therefore, a clear need, firstly, for an unambiguous screening algorithm and, secondly, for a test that is as comprehensive and complete as possible for confirmation and, in selected cases, for first-line use. Dot-line immunoassays have proven very useful for idiopathic inflammatory myopathies and systemic sclerosis, conditions in which multiple autoantibodies are relevant for a correct diagnosis [9]. We, therefore, decided to evaluate the contribution of using a comprehensive immunodot test for the liver in our laboratory, where we receive more than 2500 test requests per year. The purpose of our study was to compare the results obtained with the immunodot “10-antigen liver profile” with those obtained with the reference technique in use (IIF + ELISA). We aimed to determine the performance of this technique and assess its impact on the completeness of the examination and the time taken for diagnosis.

## 2. Materials and Methods

### 2.1. Study Design

In this retrospective study, we analysed 111 samples received at the laboratory of the Cliniques Universitaires Saint Luc, Brussels, Belgium, from June to September 2023. These samples were sent for screening or follow-up tests in the context of suspected autoimmune pathology of the liver or biliary tract, a previous diagnosis, or in the context of screening post-hepatic transplantation (for the detection of de novo AIH). No additional sampling was required, and the same serum sample provided for the routine test was used. For these samples, in addition to the ordinary tests already conducted, we performed the liver profile test with 10 antigens.

Clinical data, including comorbidities, biopsy results, and other laboratory assays, were collected for correlation with autoantibody results. The data were collected using our institutional database (Epic electronic health record).

### 2.2. Laboratory Assays

#### 2.2.1. Indirect Immunofluorescence Screening Test

The kits for indirect immunofluorescence were obtained from Inova Diagnostics and distributed by Werfen. The samples were prepared using the QUANTA-Lyser^®^ (Inova Diagnostics Inc., San Diego, CA, USA).

They consisted of sections from rat kidney, stomach, and liver for assessing, AMA, SMA, LC-1, and LKM and Hep-2 cells for ANA detection when requested. The initial dilution for ANA was 1/80 and 1/40 for AMA, SMA, LC1, and LKM, following the manufacturer’s instructions. Each run included both positive and negative antibody controls provided by the manufacturer. All analyses were performed and first interpreted with the NOVA View^®^ microscope (Inova Diagnostics Inc., San Diego, CA, USA) and then reviewed and checked by a specialized technologist and a medical supervisor.

#### 2.2.2. Confirmation Tests

Depending on the results of the screening or of our own agreement or on special request, we proceeded with a confirmatory test: Quanta lite^®^ Actin IgG, M2 EP (MIT3), LKM-1, and SLA are enzyme-linked immunosorbent assays for the quantitative detection of Ig antibodies (Inova Diagnostics Inc., San Diego, CA, USA). Normally, we perform ELISA confirmation after a positive or equivocal result with IIF. The antigens used are a purified F-actin antigen for Actin IgG, an affinity-purified recombinant antigen containing parts of PDC-E2, BCOADC-E2, and OGDC-E2 for M2, a partially purified recombinant human cytochrome P450 2D6 for LKM-1, and a partially purified recombinant human SLA antigen.

#### 2.2.3. Liver Profile 10 Ag Dot for BDI

Liver Dot kits were provided by D-tek (Mons, Belgium) and distributed by Alphadia (Mons, Belgium). The test is able to detect autoantibodies against the following antigens: M2/nPDC (subunits E1, E2, and E3 of the pyruvate dehydrogenase complex, purified bovine), M2/OGDC-E2 (subunit E2 of oxoglutarate dehydrogenase complex, recombinant), M2/BCOADC-E2 (subunit E2 of the branched-chain oxoacid dehydrogenase complex, recombinant), M2/PDC-E2 (subunit E2 of the pyruvate dehydrogenase complex, recombinant), gp210 (recombinant), sp100 (recombinant), LKM1 (formiminotransferase cyclodeaminase, recombinant), LC1 (cytochrome oxidase P450 2D6, recombinant), SLA (purified rat), and F-actin (in vitro polymerized actin filaments, purified rabbit). The test is based on the principle of an enzyme immunoassay. Test strips are composed of a membrane fixed on a specific plastic support. The strips are first incubated with patients’ sera. Human antibodies, if present, bind to the corresponding specific antigen on the membrane. Unbound or excess antibodies are then removed by washing. Upon further incubation with conjugated goat antibodies against human IgG, the enzyme conjugate binds to the antigen–antibody complexes. After the removal of the excess conjugate, the strips are finally incubated in a substrate solution. If enzyme activity is present, purple dots develop on the membrane pads, and the intensity of coloration is directly proportional to the number of antibodies present in the sample. Each strip contains two controls: the reaction control for the validity of the test and the cut-off control necessary for the qualitative interpretation of the test. All steps were performed using the BlueDiver I instrument (D-tek, Mons, Belgium) (Figure 1).

The strips are read using the BlueDiver scanner and the DrDOT software (version 4.13), which also gives us a semi-quantitative evaluation of the result (from zero to 100). Signal strength >10 arbitrary units is considered positive, and a score between 5 and 10 is considered borderline, as recommended by the manufacturer (Figure 2). To ensure intra-observer agreement, the results were checked by two independent operators.

#### 2.2.4. Statistical Analysis and Curve Fittings

GraphPad version 4.0.3 (GraphPad Prism Software, San Diego, CA, USA) was used for data analysis. Data were expressed as median (range) or n (%), as appropriate. Diagnostic performance was analysed using the following indices: sensitivity (SE) = true positives/(true positives + false negatives), specificity (SP) = true negatives/(true negatives + false positives), positive predictive value (PPV) = true positives/(true positives + false positives), negative predictive value (NPV) = true negatives/(true negatives + false negatives), and accuracy = (true positives + true negatives)/(true negatives + true positives + false positives + false negatives). Statistical significance between groups (Dot vs. IFI+ELISA) was assessed using McNemar’s Chi-squared test with continuity correction for categorical variables. A *p*-value <0.05 was considered statistically significant.

## 3. Results

### 3.1. Study Population

Out of the 111 selected patients, 61 were positive on triple tissue screening, and 50 were negative. Within this group, more than 70% were women. The mean age was 37 years (Table 1).

For all patients, toxic or infectious hepatitis (HAV, HBV, HCV, HEV, CMV, and EBV) had been previously excluded.

Among this group, twenty-nine patients had histologically confirmed autoimmune hepatitis (26%), sixteen patients had PBC (14%), and seven (6.3%) had overlap syndrome (AIH combined with PBC). 

Similarly to many other autoimmune conditions, autoimmune liver diseases are associated with a variety of other illnesses thought to have an autoimmune pathogenesis [4,9,10]. Notably, 30.6% of our total study population and 48.8% of our positive population presented with another autoimmune/autoinflammatory disease, the most frequent being autoimmune thyroiditis and diabetes (Table 2).

### 3.2. Comparison of the “Gold Standard” with Immunodots

Out of the 61 positive triple-tissue screening tests routinely performed, 33 were confirmed using ELISA (54.1%): four were LKM, eleven were M2, and eighteen were F-actin. All but two results were consistent with the patient’s clinical history, laboratory, and histology findings. We then compared the combined positive IIF + ELISA results with those obtained with the immunodot method (Table 3), and no significant difference was observed (*p*-value = 0.184).

All positive results were confirmed with the dot panel except for two samples. As far as antibodies to LC1 and SLA are concerned, we do not have a confirmatory test, and they are therefore not included in this table, which will be discussed later. Perfect agreement between the IIF+ELISA and immunodot methods was observed for AMA/M2 and LKM. As far as SMA/F-actin is concerned, by comparing the IIF+ELISA combined technique with immunodot, we theoretically lose two actin-positive samples. However, neither of the two patients was diagnosed with AIH based on biopsy and other tests, so the positive result obtained with IIF and ELISA is highly questionable. It will be interesting to follow these two patients over time.

### 3.3. Immunodot Positive with Negative IIF Screening

On the other hand, when we compared the negative IIF screening results with those from the immunodot method, we noticed that there were 23 discrepant results (Table 4).

For these samples, we successively performed confirmatory ELISA tests or ANA IIF, if not already performed, emphasizing the fact that these tests would not have been performed routinely if not requested by the clinician or specifically added by the laboratory.

By performing the immunodot test, we were able to detect the additional presence of the following autoantibodies, sometimes more than one type present in the same sample: 

Three anti-LC1: one of these was masked by the presence of anti-LKM antibodies with the IIF method. All of them have been confirmed via ELISA and matched with the diagnosis.

One positive anti-LKM: in a patient who had already been diagnosed with AIH type 2 with anti-LKM antibodies and had been on therapy since 2018, the IIF was completely negative.

Nine positive anti-sp100: for those patients, only four samples had a simultaneous ANA screening result, while for the others, it was not routinely requested and therefore not performed. We retrospectively performed an IIF on Hep2 cells to highlight the eventual presence of nuclear dots; all but one were positive. For all of them, there was a diagnosis of PBC, and three of them had a known presence of anti-mitochondria autoantibodies.

Nine F-actin positive or borderline: all confirmed retrospectively via ELISA, and six of which corresponded to patients with a diagnosis of AIH type 1. For the other three cases, it will be necessary to follow up over time to determine whether the result is significant or not.

An SLA-positive sample (also sp100 positive) was confirmed via ELISA, with two anti-gp210, both masked by the presence of anti-mitochondrial antibodies on Hep2 cells.

All immunofluorescence images were reviewed in light of these results, and we were able to confirm our initial observations.

To summarise our results, we noticed that with the dot-blot as a first approach, we would have lost two samples positive for anti-F-actin, but we recovered nine additional cases, as well as cases of anti-LKM, anti-SLA, anti-sp100, and anti-gp210 (Figure 3).

## 4. Discussion

The present study confirms the good overall performance of the 10-antigen immunodot panel in confirming the results of the gold standard test in use (IIF+ELISA). Moreover, the blot panel can provide important diagnostic information when the current routine diagnostic approach fails to detect the presence of certain autoantibodies. In our study, we found a significant number of disagreements in favour of the dot-blot test as a first-line test. This could be explained using several factors. First, the reading and interpretation of IIF images are obviously linked to the operator who performs it. For example, AMA IIF may be confused with other cytoplasmic antibodies [10,11] that are not directly associated with PBC. Some false positives may be, in fact, misinterpretations. A second review of the images should always be performed, either by a second operator or by a medical supervisor. This check is not always carried out in all the laboratories, mostly due to time or staffing constraints. Despite these measures, there will always remain a subtle difference in sensitivity and perception among operators, even if they are properly trained and experienced.

A second point of discussion could be around the search for sp100 and gp210 antibodies: over the last 20 years, several reports have described the correlation of PBC-specific ANAs with more severe disease and worse outcomes [11,12]. The importance of early detection of these antibodies is therefore clear, and their presence can induce much more rapid and severe progression of biliary pathology [13,14]. Finding anti-sp100 and anti-sp210 is not always easy as they are sometimes not visible on Hep2 cells or because the ANA visualisation can be hindered by the AMA presence or other ANA specificities, often seen in patients with the rheumatic disease [15]. In an interesting study, Invernizzi et al. demonstrated reactivity to NPC (nuclear pore complex, of which gp210 is a component) via immunoblotting in 22% of patients known to be ANA-negative via IIF [14]. In our series, five of the samples positive for sp100 or gp210 were equally positive for anti-mitochondria: the presence of the latter masked the other antibodies. Two samples, however, were completely negative in a retrospective control on Hep2 cells. This clearly shows that IIF is not ideal for the search for these autoantibodies, which is necessary for correct prognosis. In a study by Villalta et al., they also had a higher positivity rate with a line blot compared to the corresponding ANA pattern positivity observed with the IIF technique [16].

A separate argument justifying the discrepancies found when comparing the two methods is the fact that ANAs are not systematically requested by clinicians or by external laboratories that send us samples: in this case, there is no opportunity to even suspect the presence of autoantibodies such as anti-gp210 and/or anti-sp100. With the immunofluorescence technique, anti-LC1 stains hepatocytes but spares the centrilobular areas of the liver; by contrast, anti-LKM1 stains hepatocytes throughout the lobule. When both antibodies are present, anti-LKM covers the areas of anti-LC1 that are not stained. That is why anti-LC1 was “invisible” in one of our samples using the IIF technique. As for anti-SLA antibodies, they are not visible with the IIF method and are eventually sought only in the absence of other positive results when there is a very strong suspicion of hepatic autoimmune pathology. Consequently, there is a risk of missing a diagnosis or wasting precious time.

Lastly, using the immunodot technique, we recovered several F-actin-positive or borderline patients who were IIF-negative. How should we interpret these results? Are they false positives? But, upon further investigation of those patients’ files, we found that six out of nine of these reports were corroborated by clinical and laboratory history and other evidence, such as biopsy and response to therapy, and had a diagnosis of AIH 1. A possible explanation for this apparent increased sensitivity is provided by the test construction technique itself (Figure 4): F-actin is an in-house preparation consisting of in vitro polymerisation of G-actin using a specific polymerisation buffer and the addition of polymerisation-promoting elements and ATP. This F-actin is immobilised on nitrocellulose when it has reached its maximum degree of polymerisation, making the test particularly efficient.

In our study, we found almost perfect agreement between the IIF-positive samples and the dot blot for the detection of AMA, LKM, and F-actin. The only two positive samples identified via the IIF method that were not found positive via immunoblots were from patients with other diseases who had never been diagnosed with an autoimmune disease. In contrast, relying solely on the use of immunofluorescence technique, we would have missed approximately 20 positive results (Figure 5).

Even if the IIF method has long been considered the gold standard for the detection of most autoantibodies implicated in autoimmune hepatitis and autoimmune biliary disease, this technique can be time-consuming, observer-dependent, and often fails to provide information on some autoantibodies (especially in the presence of other concurrent IIF pattern) [17]. The reading of IIF slides is a constant concern, while the results obtained via immunodot are objective, as the blots are analysed using software with a well-defined cut-off [9].

Diagnosis is still challenging, and there is currently no common algorithm for the detection of autoantibodies, and this leads to variability in the management of patients [18]. This also manifests itself in variability in the way clinicians prescribe tests, and sometimes, the need arises to expand the test panel by adding different techniques to fill in gaps. This approach obviously slows down the diagnostic process and may cause clinicians to lose the overview of the patient.

The liver profile 10 Ag Dot appears to be a quick, sensitive, and specific method with a very good overall accuracy (Table 5).

Considering these promising results obtained with the dot-line test, we propose a different algorithm (Figure 6 and Figure 7) that we will reconsider over time and adjust if necessary.

We have seen that other studies [16,19,20,21] have suggested using immunodot in cases of IIF negativity but with high clinical suspicion; we propose the use of the dot technique as a confirmation test or as an alternative test in case the IIF technique is negative despite a high suspicion of autoimmune pathology. In some cases, we will propose it in the first instance as a screening test in naïve patients, with or without IIF. The only difficulty in implementing this test as a first-line diagnostic technique in our country is the fact that it is not reimbursed, and the test is charged to the patient; we hope that this obstacle can be overcome quickly in the future. On the other hand, we would like to point out that this technique is not useful in the case of follow-up. First, it provides semi-quantitative results, and second, therapy and over time may lead these autoantibodies to become negative.

However, our manuscript has some limitations. In general, this paper proposes a new diagnostic algorithm and discusses some problems with existing diagnostic methods, but there are still some shortcomings in data interpretation, consistency of results, and cost-effectiveness that need further improvement and in-depth research.

## 5. Conclusions

The choice of the best option for the detection of autoantibodies in liver/biliary tract pathologies depends on many factors, such as the centre’s expertise, the technologies available to laboratories, local prevalence, the level of diagnostic accuracy needed, and overall expenditure [10]. Nevertheless, in a specialized referral laboratory, where the prevalence of patients with liver and biliary tract autoimmune pathologies is assumed to be higher, the ability to test for all relevant antibodies simultaneously could lead to preferring multiple immunodot profiles for a first-line assay.

In the present study, we have evaluated the performance of the liver profile 10 Ag Dot (D-tek, Mons, Belgium) for the detection of autoantibodies in relation to autoimmune hepatitis and primary biliary cholangitis. Our results showed an overall sensitivity and specificity of 100% and 95.2%, respectively, compared to IIF, which showed lower sensitivity and the same specificity. Based on these results, we propose the use of this assay as confirmation of the IIF method and as a first-line test in certain situations.

## Figures and Tables

**Figure 1 diagnostics-14-01882-f001:**
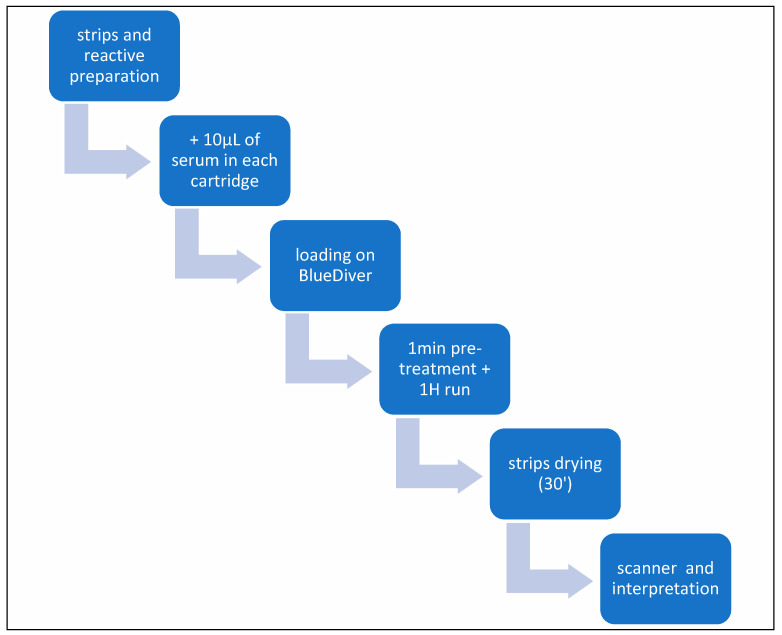
Diagram of test stages.

**Figure 2 diagnostics-14-01882-f002:**
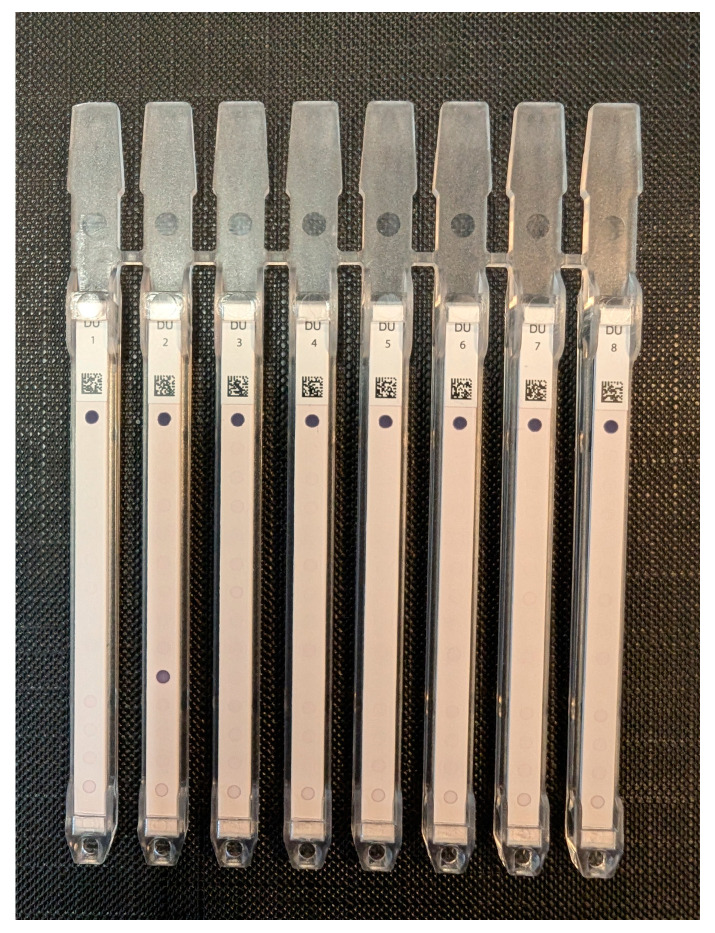
An 8-sample run; patient 2 is positive.

**Figure 3 diagnostics-14-01882-f003:**
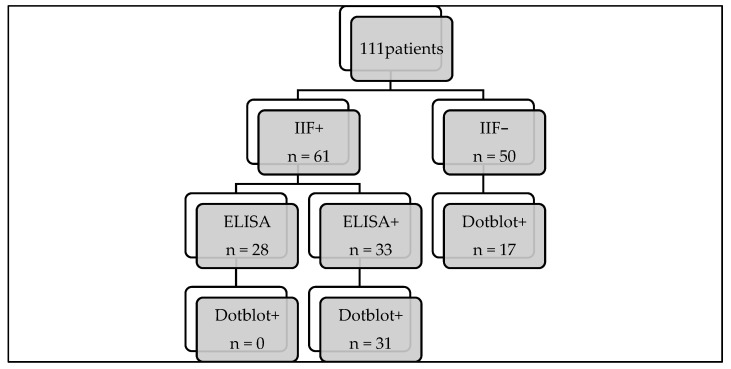
Result flowchart. IIF: indirect immunofluorescence; ELISA: enzyme-linked immunosorbent assay.

**Figure 4 diagnostics-14-01882-f004:**
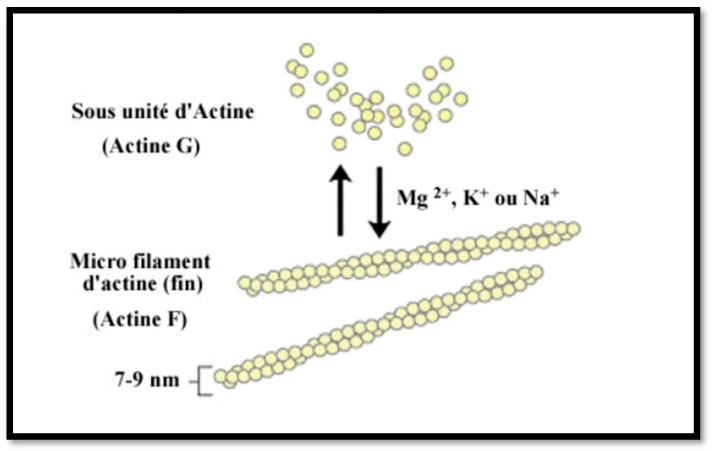
Polymerisation of G-actin. Image courtesy of D-tek (Mons, Belgium).

**Figure 5 diagnostics-14-01882-f005:**
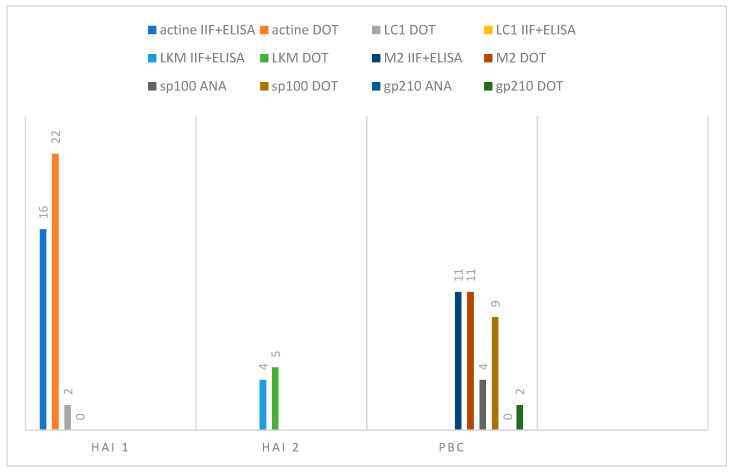
Comparison of IIF/ELISA and dot blot results in AIH1, AIH2, and PBC. IIF: indirect immunofluorescence; ELISA: enzyme-linked immunosorbent assays; AIH: autoimmune hepatitis; PBC: primary biliary cholangitis; SMA: F-actin reactive smooth muscle autoantibodies; LKM: liver kidney microsome; LC: liver cytosol; M2: mitochondria-associated autoantibody; sp100: nuclear body speckled 100KDa protein; anti-gp210: glycoprotein of the nuclear pore complex.

**Figure 6 diagnostics-14-01882-f006:**
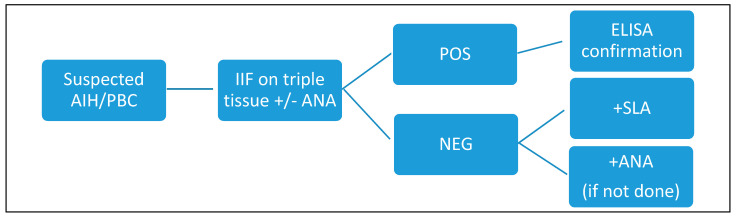
Currently followed algorithm. AIH: autoimmune hepatitis; PBC: primary biliary cholangitis; IIF: indirect immunofluorescence; ANA: antinuclear antibodies; ELISA: enzyme-linked immunosorbent assay.

**Figure 7 diagnostics-14-01882-f007:**
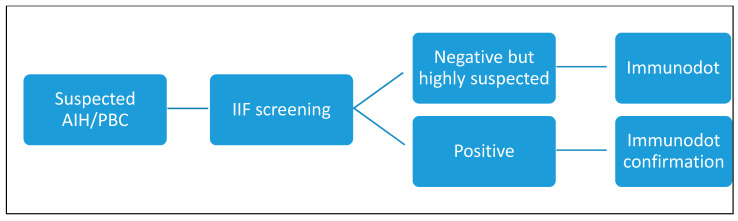
Newly proposed algorithm. AIH: autoimmune hepatitis; PBC: primary biliary cholangitis; IIF: indirect immunofluorescence.

**Table 1 diagnostics-14-01882-t001:** Patients’ characteristics.

		Number of Patients
Sex	F	79 (71%)
	M	32 (29%)
Age (mean and range)		37 (1–86)
IIF screening	Positive	61 (54.9%)
	Negative	50 (45.1%)
AIH		29 (26%)
PBC		16 (14%)
Overlap variant		7 (6.3%)

F: female; M: male; IIF: indirect immunofluorescence; AIH: autoimmune hepatitis; PBC: primary biliary cholangitis.

**Table 2 diagnostics-14-01882-t002:** Other autoimmune diseases in the total study population.

Comorbidities	N° Patients (%)
Autoimmune thyroiditis	10 (9%)
Autoimmune diabetes	8 (7.3%)
Celiac disease	5 (4.5%)
IBD	2 (1.8%)
Rheumatoid arthritis	2 (1.8%)
Sjogren syndrome	2 (1.8%)
Other	5 (4.5%)
Total	34 (30.6%)

N°: number; IBD: inflammatory bowel disease.

**Table 3 diagnostics-14-01882-t003:** Comparison of the “Gold Standard” with immunodots.

Test	Screening + ELISA	DOT
LKM	4	4 (100%)
M2 *	11	11 (100%)
F-actin	18	16 (88.9%)

* All subunits combined: M2/nPDC, M2/OGDC-E2, M2/BCOADC-E2, and M2/PDC-E2. LKM: Liver kidney microsome; M2: anti-mitochondria—autoantibodies 2.

**Table 4 diagnostics-14-01882-t004:** Immunodot positive with negative or partially negative screening.

	DOT	Screening IIF (Triple Tissue)	ANA Screening	Confirmation	Clinical Data
1	LC1: 82 (LKM+)	LKM+, LC1−	Speckled	ELISA+	AIH 2 diagnosis
2	LC1:21	Negative	Nucleolar	ELISA+	AIH 1 diagnosis
3	LC1: 22	Negative	Negative	ELISA+	AIH 1 diagnosis
4	LKM: 9	Negative	Speckled	ELISA+	LKM+ AIH 2 diagnosis
5	F-actine: 10	Negative	Speckled	ELISA+	Post-transplant for BA
6	F-actine: 8	Negative	Speckled	ELISA+	AIH+PBC diagnosis
7	F-actine: 19	Negative	Not done	ELISA+	Suspicion of AIH 1
8	F-actine: 18	Negative	Speckled	ELISA+	Hepatic cytolysis
9	F-actine: 11	Negative	Homogeneous	ELISA+	Previous diagnosis of AIH 1
10	F-actine: 87	Negative	Nucleolar	ELISA+	AIH 1 diagnosis
11	F-actine: 44	Negative	Speckled	ELISA+	AI cholangitis
12	F-actine: 16	Negative	Speckled	ELISA+	Cholestase
13	F-actine: 62	Negative	Speckled	ELISA+	AIH 1 diagnosis
14	sp100: 89 (M2+)	AMA+	Not done	ANA + (ELISA+)	PBC diagnosis
15	sp100: 99 (M2+)	AMA+	Not done	ANA + (ELISA+)	PBC diagnosis
16	sp100: 21 SLA:100	Negative	Not done	ANA + (ELISA+)	AIH+PBC diagnosis
17	gp210: 87 (M2+)	AMA+	Mitochondria	ANA− for gp210	PBC diagnosis
18	sp100: 100	Negative	Speckled+nuclear dots	ANA+	Unknown
19	sp100: 64, gp210: 53 (M2+)	AMA+	Not done	ANA + (ELISA+)	PBC diagnosis
20	sp100: 19	Negative	Speckled	ANA−	PBC diagnosis
21	sp100: 98	Negative	Nuclear dots	ANA+	PBC diagnosis
22	sp100: 100	Negative	Nuclear dots	ANA+	Previous sp100+
23	sp100: 92	Negative	Nuclear dots	ANA+	AIH diagnosis

Immunodot results: 0–5 negative, 5–10 borderline; >10 positive. BA: biliary atresia; IIF: indirect immunofluorescence; ELISA: enzyme-linked immunosorbent assays; AIH: autoimmune hepatitis; PBC: primary biliary cholangitis; LKM: Liver kidney microsome; LC1: liver cytosol; AMA: anti-mitochondria—autoantibodies; sp100: nuclear body speckled 100KDa protein; anti-gp210: glycoprotein of the nuclear pore complex.

**Table 5 diagnostics-14-01882-t005:** Overall performances of the immunodot panel compared with the IIF+ELISA method.

	Sensitivity (%)	Specificity (%)	PPV (%)	NPV (%)	Overall Accuracy (%)
Immunodot	100	95.2	100	94.4	97.3
IIF/ELISA	71.8	96.8	100	75.9	83.5

PPV: positive predictive value; NPV: negative predictive value.

## Data Availability

Data will be available upon request to the corresponding author. The data are not publicly available due to privacy restrictions.

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
