# Peer review of "Evaluation of a Ten-Antigen Immunodot Test in Autoimmune Hepatitis and Primary Biliary Cholangitis: Lessons Learned for a Tertiary Care Academic Hospital"

_diagnostics, 2024, doi:10.3390/diagnostics14171882_

Round 1
Reviewer 1 Report
Comments and Suggestions for Authors
The title is acceptable.
Abstract: It is short and informative.
Introduction: This is an informative piece with a useful background and recent references. The introduction also outlines the problem and the study's goal.
Materials and Methods
It is informative and includes the methods in a clear and detailed manner. Also, it included statistical analysis.
The results included clear figures and tables.
It could be better if authors added photos for the immunodot to show examples of positive and negative results, as well as a diagram of the procedures.
The discussion is a decent presentation.
The references section contained sufficient recent references.
Author Response
Thank you in general for your positive and constructive comments.
In details:
Comment 1: It could be better if authors added photos for the immunodot to show examples of positive and negative results, as well as a diagram of the procedures
Response 1: That'a a very good point indeed. So we've added a photo with an example of positive and negative samples, as well as a figure summarising the procedure for carrying out the dots. Changes and additions are highlighted in the article so that you can assess the corrections. The photo and diagram are also loaded separately.
Reviewer 2 Report
Comments and Suggestions for Authors
The few concern in the manuscript
clarify that the 10 immunoblot antigens test was done on 111 samples developed by you or the company. if it is a company product are they not tested on the given 10 markers? What is the main goal of the developed panel? Secondly did the company did the performance evaluation of the said kit/panel.
if the company do all the validation what is the novelty of your study.
Author Response
Thank you for your review and comments.
Comment 1: if it is a company product are they not tested on the given 10 markers? What is the main goal of the developed panel? Secondly did the company did the performance evaluation of the said kit/panel.If the company do all the validation what is the novelty of your study.
Response: This test have been created by a company and validated technically by them: they evaluated repeatability, reproducibility, analytical and sensitivity and specificity.
The novelty of our study is that the test in question is compared in a clinical context, to techniques considered gold-standard (Immunofluorescence and ELISA) and demonstrates its non-inferiority. Moreover, this technique identified over 20 additional autoantibodies in samples initially negative by routine immunofluorescence technique and the vast majority of which turned out to be true cases of autoimmune disease of the liver or biliary tract.
Those results are therefore so promising to consent us to propose in our teaching hospital a new diagnostic algorithm for the detection of autoantibodies in those autoimmune pathologies, which could lead to more rapid and targeted treatment by clinicians.
Finally, no clinical evaluation of this complete kit and its place in the diagnostic process is yet available in the literature.
Thank you for your time for this review
Reviewer 3 Report
Comments and Suggestions for Authors
The manuscript entitled " Evaluation of a ten immunodot antigens test in autoimmune hepatitis and primary biliary cholangitis, lessons learned for a tertiary care academic hospital.” describes the efficacy and usage of a dot blot 10-parameter test in autoimmune liver diseases, autoimmune hepatitis (AIH) and primary biliary cholangitis (PBC).
Authors clearly demonstrate the utility of this immunoblot test, concluding in a proposal of a new algorithm in the procedure of diagnostic testing for these diseases.
The group of the patient used to evaluate the test is acceptable to support the conclusions of the study. Similar studies that conclude in the useage of a dotblot test for AIH and PBC diagnosis have been already published, but only two or three in the last 2 years. The new information that this study can communicate to the science community is the proven reliability of the certain commercial kit that was used in the study.
Overall, is a well written paper, I don’t have to mention specific points in the manuscript.
Authors describe a good- practice guide for the screening of autoimmune liver diseases based on their hand-on-bench work that can be applied in laboratories that are called to deal these diagnostics tests in their routine.
In Table 4, it would be more explainable if you could quote the ANA patterns that was observed on IIF for these samples that are discussed in details?
Author Response
Thank you very much for your encouraging and positive comments. I take it from these that we managed to get the message across, thank you.
In details:
Comment 1: In Table 4, it would be more explainable if you could quote the ANA patterns that was observed on IIF for these samples that are discussed in details?
Response: Actually an interesting figure to specify: we added it to table 4; thank you
Round 2
Reviewer 2 Report
Comments and Suggestions for Authors
ok